# The Complementary Role of Gestures in Spotted Hyena (*Crocuta crocuta*) Communication

**DOI:** 10.3390/ani15101366

**Published:** 2025-05-09

**Authors:** Andrew J. Laurita, Stephanie A. Poindexter

**Affiliations:** 1Environment and Sustainability Department, University at Buffalo, Buffalo, NY 14068, USA; sapoinde@buffalo.edu; 2Anthropology Department, University at Buffalo, Buffalo, NY 14261, USA

**Keywords:** spotted hyena, gestural communication, behavioral innovation, fission–fusion social societies, visual communication

## Abstract

Spotted hyenas live in fission–fusion societies, which require individuals to have robust social memory to congregate effectively in gregarious settings. Their sophisticated social structure confers adaptive benefits to individuals that can quickly convey their dominance rank and recognize the social statuses of others. Here, we focus on the role of visual communication (focusing on gestural communication) in this capacity and how it complements olfactory and acoustic communication in mediating conspecific communications in spotted hyenas. Facial and manual gestures aid spotted hyenas in maintaining relative dominance, mitigating the likelihood of physical altercations, and promoting group cohesion and cooperation. We examine how spotted hyenas use gestural communication in captive and wild settings and investigate how captivity modifies specific gestural behaviors.

## 1. Introduction

Spotted hyenas (*Crocuta crocuta*) live in a fission–fusion society, where individuals switch back and forth between living gregariously and solitarily [1]. Individuals congregate and cooperate to defend kill sites from intraspecific and interspecific competitors while defending territory from opposing clans and occasionally for group mating [2]. However, spotted hyenas primarily live alone or in small subgroups (i.e., aggregations exceeding five individuals are rarely observed outside clan territorial skirmishes, communal den congregations, and while hunting unusually large prey) [3]. Following group “fission”, most spotted hyenas move (7–8 mph ‘lope’, or a comfortable travel speed to ~36 mph ‘gallop’, their peak velocity) between varied habitats (i.e., individuals not part of the dominant matriline), including open savannahs, shrubby sahels, and densely wooded forests [4]. To successfully coordinate, individuals must be able to send and receive signals that vary across spatial and temporal scales and social contexts. Each habitat has different potential environmental obstacles that affect long-distance communications (e.g., increased potential prey eavesdropping in open savannahs and increased attenuation from trees and foliage in forests) [5,6].

Unique environmental constraints shaped the development of the linear matriarchal hierarchy and advanced social intelligence in spotted hyenas. Extreme resource limitation likely selected for gregariousness in spotted hyenas, enabling them to acquire and defend food cooperatively, and for behavioral plasticity, which allows migration between clans when advantageous [7]. Intensive feeding competition and protracted skull development required to exploit a feeding niche permitting bone consumption favored the evolution of large, aggressive “role-reversed” females capable of displacing male competitors to provide additional food provisions to their unusually altricial cubs (infants with a need for parental care investment) [7]. Enhanced aggressiveness further allows females to dominate male conspecifics at feeding sites, permitting them greater access to food resources to share with their kin [8]. These traits led to a strict matriarchal system in which philopatric females inherit dominant ranks from their mothers. In contrast, males acquire relative dominance over other males (though all males are socially submissive to all females) by duration in a clan (i.e., the longer a male has lived in a clan, the more dominant he is) [9,10]. Male social rank ascension is almost always linear. However, females occasionally attempt to usurp a matriarch, and success primarily depends upon the number of individuals in the coup compared to those allied with the established matriline [9,10,11,12].

Rank acquisition begins as early as a week after birth, with associative learning dictating this process as aggression from cubs towards lower-ranking mothers is reinforced while aggression directed towards higher-ranked mothers is extinguished [13]. This social system requires individuals to recognize third-party relationships (a rare skill among animals requiring high cognitive abilities), memorize individual ranks, and display nepotistic behavior (allocating additional provisioning to kin and/or other conspecifics of greater relatedness to the provider compared to less related conspecifics) to enhance their fitness indirectly [8,12]. Social rank determines food access priority, potential cub-rearing aid, and coalitionary support during inter-clan conflicts. Observations of wild spotted hyenas have suggested that associations between female conspecifics align with kin selection theory. This states that cooperation is greatly enhanced between related individuals, reducing the likelihood of aggressive behaviors between conspecifics of greater relatedness values [13,14]. Memorizing and conveying information about social rank through signals associated with behaviors such as nepotistic favoritism requires a large cognitive capacity and the ability to recognize and emit complex signals (n ~75–95) that are plastic to variable social situations [8,15]. The ‘whoop call’ has variable forms (‘slow bouts’ and ‘fast bouts’) emitted for different purposes between adult males and females. Males appear to use these vocalizations for self-advertisement (through almost exclusive employment of spontaneous, ‘slow bouts’) to male conspecific competitors and prospective mates. Females express a mix of “whoop call” types during heightened arousal, displacing socially inferior conspecifics and/or reminding conspecifics of their respective social ranks and recruiting clanmates for territory and kill site defense [2,16,17].

Spotted hyenas’ behavior, social structure, and habitat demand that multimodal communications (signals that comprise two or more sensory modalities, such as visual and acoustic) be employed to ensure signal efficacy and accuracy [18]. Visual signals are adaptively employed to complement the role of olfactory and acoustic signals. Despite this, across 46 peer-reviewed articles centered on spotted hyena social cognition and communication (see Appendix A), specific communication signal mechanisms, and cooperative tactics, we found that only 14% focused on vision. Here, we employ a novel perspective in synthesizing and contextualizing the significance of gestures (a type of visual communication) in spotted hyena communication and social interactions to offer new insight into the relevance of this communication modality in upholding their social hierarchy. For the first time, we further examine how captive and wild populations differ in their implementation of all communication modalities in mediating social interactions, though we primarily focus on gestures. This will allow us to rectify a significant knowledge gap, as only 18% of the literature reviewed here has studied captive populations. However, we have found no evidence of comparisons between captive and wild populations’ gestural communication usage.

### Review Objectives

Objective 1: Reveal new insights into gestural communication’s role in spotted hyenas’ social communication, especially in relation to acoustic and olfactory communications.Objective 2: Compare captive and wild spotted hyena gestural signal repertoires to show how these signals’ form, frequency, and function vary between these populations.

## 2. Gestures in Spotted Hyenas

Gestural communication is a form of visual communication used in variable light conditions, acting over short distances, greatly affected by the environment, weakening signal efficacy [19]. Gestures are a communicative method or aid in which a visible, symbolic action conveys information that has a mutually understood definition between the sender and receiver [19]. Gestures require the participation of a receiver to attend to a signal, meaning gestures are inconspicuous unless the receiver(s) are visually attentive to the sender [19]. Numerous studies have identified gestural communication across diverse taxa, including various bird species, social primates, and canines [19,20,21].

Spotted hyenas understand visual signal optimization by assuring the receiver can attend to an emitted signal and alter signal employment for variable social contexts (i.e., through an audience effect based on the relative social rank of the receiver) [19,22]. Most gestures in the spotted hyena repertoire function as redundant signals (signals conveyed alongside other gestures or modalities that express the same information). For example, hyenas often combine ‘flattened ears’ with a ‘head bob’ to signal submissive intent during agonistic interactions. They also pair gestures with acoustic or olfactory signals to reinforce meaning [6,8,23]. Some of the most common conspecific gestures spotted hyenas use include changing positions of the body, ear, clitoris, penis, and tail, performed most often during dominance interactions and at social reunions [6]. However, researchers have identified more distinct, specialized gestures (Table 1). Due to the rigidity of the spotted hyena social structure, most specialized signals have evolved to effectively convey varying levels of submission and/or dominance displays [8,23]. For example, both ‘ear flattens’ and ‘grin and retreat’ convey submission, but the latter conveys this more strongly than the former [10,23]. This is akin to the submissive ‘appeasement grin’ vs. the extremely submissive ‘crouching posture’ observed in gray wolves [24]. The other predominant function of specific gestures is to convey emotional states. Spotted hyenas exemplify this when they contagiously yawn in group settings, mirroring conspecifics’ imminent behavioral state changes. They use this information to decide which social partners to associate with [25].

Spotted hyenas employ two types of gestures: (1) facial (gestures that involve facial movements) and (2) manual gestures (gestures that involve body movements, sometimes referred to as “body signals”) to mediate social interactions [23,25] (Table 1). Researchers have observed two facial gesture types. “Head-bobbing” (HB) and “relaxed open mouth” (ROM) [23]. ROM is observed more frequently than HB by low-ranked and high-ranked individuals. Still, the difference is less for low-ranked individuals (~40% HB, 60% ROM for low-ranked individuals; ~15% HB, 85% ROM for high-ranked individuals) [12,23,25]. HB is an extreme form of appeasement and is mainly exhibited by cubs directed towards their mothers or other cubs, by males directed towards females, by males directed towards dominant males, and by subordinate females as an act of reconciliation from agonistic encounters with dominant females [23,26]. HB can also initiate social affiliative acts, often by subordinate females towards dominant females [23]. ROM is an anticipatory signal that is consistently highly redundant in frequency and duration to maximize the likelihood that actions performed in play are not mistaken for agonistic behavior [23,26,27]. ROM intends to convey to the play partner that the sender’s proceeding actions (e.g., biting, nipping) are for play only [18,24]. ROM is a valuable gesture in the spotted hyena cub’s repertoire since play interactions form the bulk of cub-to-cub interactions. Play is helpful in modulating aggression and facilitating integration into the clan by aiding in establishing dominance hierarchies [27]. ROM is not unique to spotted hyenas in either form or function; similar gestures occur in species such as ring-tailed lemurs (*Lemur catta*), coyotes (*Canis latrans*), and South American sea lions (*Otaria flavescens*) [20,23,28,29,30]. It is highly redundant in frequency and duration to maximize the likelihood that actions performed in play are not mistaken for agonistic behavior [24,26]. ROM is a valuable gesture in the spotted hyena cub’s repertoire since play interactions form the bulk of cub-to-cub interactions. Play is helpful in modulating aggression and facilitating integration into the clan by aiding in establishing dominance hierarchies [27].

## 3. How Gestural Communication Complements Acoustic Communication

Spotted hyena vocalizations have evolved to be effective for a fission–fusion society characterized by sophisticated social networks and relationships. Possessing a diverse repertoire of vocalizations, spotted hyenas can employ fine-tuned calls for variable spatial and temporal conditions, accounting for the recipient’s social relationship to the sender and adjusting for environmental variability in sound transmission (e.g., employing a low-frequency ‘whoop’ vocalization in densely wooded habitats where environmental attenuation is high) [6,31]. Acoustic communication holds particular value for spotted hyenas because they can emit these signals rapidly during high-tension contexts, such as recruitment calls at kill sites and clan wars [17,31]. For instance, matriarchs or other socially dominant females can emit recruitment ‘whoop’ calls comprising 5–8 low-frequency signals intended to elicit an immediate response from conspecifics up to 50 km away [17,31]. This is necessary to bolster clan numbers as quickly as possible (i.e., the distance a receiver is from the sender dictates this as the recruitment whoop call elicits responses within <10 s post-emittance) to intimidate rival clans and/or interspecific competitors, such as African lions (*Panthera leo*) [11,17,31].

During gregarious interactions, gestural communication helps maintain complex relationships in the spotted hyena’s sophisticated social hierarchical system [23,32]. Gestural communication permits improved spatial resolution compared with acoustic communication, which is more fine-tuned for temporal resolution [33,34]. Though typically compromised, this enables gestural communication to be useful for essential functions such as prey identification and conspecific recognition [33,34]. Spotted hyenas preferentially use gestural signals to convey important information during close-range social interactions, such as conciliatory head and penis rubbing [6,23,35].

Due to being primarily active at night and during crepuscular hours, spotted hyenas have vision attuned to accentuating sensitivity rather than resolution [34]. Their feeding ecology selects forward-facing eyes that maximize depth perception and facilitate more effective hunting [34]. Spotted hyenas employ conspicuous gestural movements, including ear, head, and tail movements, to help coordinate hunts during nocturnal hours, an optimal time to hunt to avoid daytime heat and competition from African lions [6,34]. Spotted hyena gestures are akin to the white throat badge of the eagle owl (*Bubo bubo*), in which the brightness of this badge aids in facilitating nocturnal mating and territorial displays [36]. These physiological adaptations support conspecific communication at night, where gestural signals often accompany acoustic signals redundantly. For example, hyenas may use acoustic contact calls to maintain cohesion among small, mobile subgroups while pairing them with gestures to increase the likelihood of successful signal transmission and reduce the risk of a failed hunt due to miscommunication in tactical coordination [8,23,34].

## 4. How Gestural Communication Complements Olfactory Communication

Olfactory communication in spotted hyenas broadly serves two vital functions: (1) it allows for a persistent signal that can effectively communicate individual sex, age, reproductive status, and identity to conspecifics, and (2) it allows territory marking that can be tracked and maintained over a broad and fluctuating temporal and spatial scale [6,37]. Spotted hyenas exhibit a behavior known as pasting (defecating in tactically placed latrines to reinforce scent markings) that conspecifics (within and outside the sender’s clan) can use to glean social and health identification information about the sender for up to 40 days following deposition [4]. Conversely, gray wolves leave scent marks that persist for 2–3 weeks [38]. In spotted hyenas, this supports the accessibility of social status information (as spotted hyenas possess unique scent profiles) and facilitates their fission–fusion society characterized by frequent variable individual locomotor patterns within and between clan territories [39]. Spotted hyenas adjust their locomotor patterns over varying timescales to accommodate seasonal variations in prey density incurred by hunting migratory prey [39,40]. This lifestyle benefits from using olfactory signals as they diffuse slowly [34,37]. For example, spotted hyenas employ scent-marking in preparation for hunting zebras (*Equus quagga*), a behavior unobserved when hunting other, less migratory species [40]. Furthermore, low-ranking females and males are especially attentive to olfactory signals, as they frequently must travel to remote feeding sites for sustenance and/or breeding opportunities [41,42]. Recognizing and appropriately responding to olfactory signals can confer adaptive benefits by conserving energy expenditure costs from excessive locomotion [1,41].

Spotted hyenas use acoustic and gestural signals to transmit information more rapidly than olfactory signals. Still, gestural signals offer a significant advantage in this role by conveying behavioral states more efficiently. For example, a single ‘head bob’ can communicate submissive intent to a conspecific. Conversely, conveying the same message acoustically would require a more energetically costly A and T whoop call bout [1,23,34]. As such, gestural signals are valuable for energetically inexpensive direct communications at relatively quick rates in redundantly accompanying olfactory signals.

## 5. Social Communication in the Wild vs. Captivity

Wild and captive populations approach social learning through similar mechanisms and exploit opportunity benefits similarly [43]. Rank acquisition and application of this knowledge for mediating social interactions, cooperative problem-solving approaches, and nepotistic behaviors appear to be effectively unchanged between captive and wild populations [43,44]. Limited resources and intense competition degrade opportunities for lower-ranked individuals to develop long-lasting affiliative relationships with dominant conspecifics in wild populations [43,44,45]. In captive settings, resources are relatively abundant compared to their wild counterparts, and intensive feeding competition is comparatively reduced, thus loosening the strict adherence to the social hierarchy observed in wild populations [6,44]. This broadens the potential for captive individuals to exhibit more exploratory behavior (which could be risky in the wild) and develop innovative behaviors, including a diverse communication repertoire (e.g., employing more tonal, prolonged ‘groan’ vocalizations to elicit attention from conspecifics due to being more conspicuous than less tonal ‘groans’. The potential consequence of this in wild conspecifics appears vulnerable due to its association with attempting to elicit supplementary provisional allocations necessitated by an inability to acquire provisions, a concern less necessary in captive environments) [1,6,44,45]. For instance, as evidence suggests, spotted hyenas learn via localized stimulus enhancement (observers are most attentive to specific elements of stimulus they observed being manipulated by a demonstrator), being more willing to explore novel objects in conjunction with not needing to allocate as much time acquiring provisions as wild conspecifics, permits captive spotted hyenas to more willingly interact with a puzzle box that may confer enrichment benefits, encouraging them to subsequently interact with it in the future and enhance their ability to solve the puzzle box [44,45]. However, multiple studies have demonstrated that social learning opportunities have negligible effects on the success of completing problem-solving tasks in captive populations, but rather, the reduction of neophobic behaviors (fear of unfamiliar stimuli) in these populations significantly improved their problem-solving abilities [43,44,45]. In a comparative novel puzzle box task between 32 captive (M: 15, F: 17) and 30 wild spotted hyenas (M: 14, F: 16), only 14.5% of wild spotted hyenas completed the task on their first attempt, whereas 73.7% of captive individuals were successful in in their first attempt at task completion [45]. Captive spotted hyenas likely showed this result because they exhibit reduced neophobic tendencies compared to their wild conspecifics, rather than because they possess superior problem-solving abilities gained through social learning [43,45]. Successful spotted hyenas (wild and captive) exhibited diverse exploratory behavior compared to unsuccessful subjects, and captive individuals were significantly more consistent in displaying diverse exploratory behavior than wild individuals [4]. Researchers also found that neophobic behavior significantly inhibited successful task completion, as the absence of neophobic tendencies significantly correlated with initial task completion success [45].

Captive spotted hyenas show greater functional plasticity in their gestural repertoire than their wild counterparts. They interact within smaller spatial scales and more frequently with the same conspecifics, which may promote rapid individual or group-level variation in dyadic interactions. For instance, reduced space decreases reliance on energetically costly long-distance vocalizations. As a result, hyenas may use short-range vocalizations more frequently and adapt them for new functions, such as dominance displays [6,43,44]. To our knowledge, captive spotted hyenas do not exhibit any novel gestures unobserved in their wild counterparts; they exhibit modified forms of shared gestures that are unique (Table 2) [1,6,23,44,45]. As spotted hyenas live in a fission–fusion social system, they likely acquire and refine their gestural repertoire by the exact mechanism as apes living in fission–fusion societies—defined by the revised social negotiation hypothesis (RSNH) [46]. RSNH posits that individuals continuously shape gestures within dyadic interactions, modifying innate gesture forms in idiosyncratic ways to suit varied contexts [46]. Redundant gestures help overcome dyadic social rank disparities by conspicuously coordinating turn-taking between conspecifics during hunts [44,46]. Captive individuals with abundant resources may co-opt redundant gestures into new roles, such as emphasizing their intentional state [15,23,44,45]. Gestural signals are optimized for rich spatial resolution at short distances, thus providing more potential opportunities for behavioral innovations in captive populations where individuals coexist in gregarious settings more often than wild conspecifics due to the spatial confines of artificial enclosures [15,23,34]. A lack of intellectual stimulation may inhibit captive individuals from developing behavioral innovations, leading to cognitive degradation [15,43,44,47].

## 6. Conclusions

Spotted hyenas use acoustic, olfactory, and gestural signals in complementary ways to guide social interactions and maintain group cohesion [15]. While olfaction supports communication across delayed spatial and temporal contexts, individuals preferentially use acoustic signals when high temporal resolution is required [6,33,34]. Individuals rely on gestural communication as a key component of social interactions, using it for its superior spatial resolution compared to acoustic signals and faster deployment than olfactory signals. For example, the ‘forward thrust’ gesture is more effective as an intimidation tactic when employed by larger coalitions, as individuals take turns in “waves” pressing forward and then receding to emphasize their numbers [9,23]. Indeed, clans with more members often employ more “thrusts” in this display during clan wars, emphasizing the heightened boldness of the clan with the numeric advantage [9,11,23]. This is akin to ‘pant-hoot’ vocalizations observed during territorial affronts between neighboring wild chimpanzee communities in which communities with a greater numeric advantage will exhibit more vocal ‘pant-hoots’ [48]. Gestures also redundantly enhance the conspicuousness of acoustic and olfactory signals [23,34]. This may be particularly relevant for captive spotted hyena populations due to their confined space and reduced resource competition compared to wild populations. Indeed, as evidence suggests that captive populations possess the ability to innovate upon the form of innate gestures observed in wild populations, yet not necessarily develop novel gestures, we suggest that reduced neophobia, increased forced social interactions, and greater resource availability leads to more diverse forms of gestures but fewer total gestural signals in captive populations compared to wild populations (as several gesture types, such as ‘crawl approach,’ are reserved for fusion interactions and food contexts) [43,44,45]. It warrants further study into how gestural communication differs in captive populations from wild conspecifics. We highlight a persistent oversight in treating gestures as secondary in the communication systems of spotted hyenas. In reality, gestural signals enhance access to social information and support coordination during critical activities such as clan territorial conflicts, hunting, and avoiding physical altercations [6,15,23]. Furthermore, we contend that the role of gestural communication in maintaining social cohesion is vital, as is evidenced in the preservation and subsequent innovation of many gestures in captive populations, along with captive populations using gestures to aid in cooperative tasks [15,23,44].

## Figures and Tables

**Table 1 animals-15-01366-t001:** Spotted hyena ethogram depicting all recorded distinct gestures and their primary functions. **Bolded** gestures indicate social rank signaling, underlined gestures reflect emotional expression, and *italicized* gestures occur in other contexts (see descriptions for details).

Spotted Hyena (*Crocuta crocuta*) Gestural Repertoire		
Behavior Name	Gesture Type	Description
**Head bobbing ^4^**	Facial	The subject avoids eye contact and lowers the entire cephalic region while pulling the ears back toward the neck. This can then be followed by moving the head back up to a neutral position and repeating the lowering motion, but not always (shows submission).
**Bite shakes ^1^**	Manual (Head)	The subject begins to gnash their teeth in the air repeatedly in a chomping motion while simultaneously moving their head back and forth laterally at a vigorous pace (conveying extreme aggression).
**Ear flattens ^1^**	Manual (Head)	The subject avoids or intermittently makes eye contact, never directly, and pulls the ears back toward the neck as a submissive signal.
**Back away ^1^**	Manual (Body)	The subject moves away from the receiver at a normal walking pace (normal gait or slow lope), but the head is directed perpendicularly to the intended receiver (submissive signal).
**Grin and retreat ^1^**	Manual (Body)	The subject quickly performs a facial movement in which the upper lips are widened, and the lower lips appear at an elevated angle. Then, the subject hastily turns the entire body away from the receiver and physically relocates to another spatial area at a fast transverse gallop. No direct contact is observed in any phase (extreme submissive signal).
**Crawl approach ^2^**	Manual (Body)	The subject’s hind legs are bent, ears flattened, mouth slightly open, and tail straight up or bent forward as it moves towards the receiver at a slow lope (extreme submissive signal).
** Affiliative greeting ^2^ **	Manual (Body)	The subject moves towards the receiver. Eye contact is held with the head at a lateral angle if the receiver is socially dominant, or eye contact is held with the head facing forward to a socially subordinate receiver; then, the subject lightly brushes their head against the receiver’s head and/or their genital regions against those of the receiver (in either case, the phallus is erect).
** Relaxed open mouth ^4^ **	Facial	Subject eye contact is maintained, ears are straight up or slightly tilted towards the sides of the head, and the upper mouth is opened widely. Canines do not protrude beyond normal, giving the sender a pointed snout appearance (initiates play).
** Spontaneous yawn ^5^ **	Facial	The subject opens its mouth, sometimes protruding its tongue, while simultaneously inhaling deeply until the mouth opening reaches the acme, which exposes the teeth. Mouth closing and air exhalation are more rapid than the mouth opening and inhalation phases.
*Foot spurring* ^3^	Manual (Body)	Subject paws ground in a back-and-forth motion with their foot at a slow pace while a receiver is attending to them. Individuals use this as a de-escalatory or anticipatory signal of agonistic behavior.
*Forward thrust* ^3^	Manual (Body)	The subject quickly lurches their entire body toward a receiver, then recedes at an equivalent rate. This may be repeated several times in succession, but not constantly (used as an intimidation signal).

^1^. Hayssen & Noonan, 2021 [6]; ^2^. Boydston et al., 2001 [9]; ^3^. Drea et al., 1996 [10]; ^4^. Nolfo et al., 2022 [23]; ^5^. Casetta et al., 2022 [25].

**Table 2 animals-15-01366-t002:** Spotted hyena ethogram depicting how each gesture type is observed between captive and wild populations. We found no evidence of a defined gesture exclusively observed in captive populations.

	Gesture
Observed in captive and wild spotted hyenas with identical forms	Head bobbing ^4^ Relaxed open mouth ^4^ Spontaneous yawn ^5^ Bite shakes ^1^ Ear flattens ^1^ Back away ^1^ Grin and retreat ^1^
Only observed in the wild	Crawl approach ^2^
Gestures with differing forms	Affiliative greeting ^2^ Foot spurring ^3^ Forward thrust ^3^

^1^. Hayssen & Noonan, 2021 [6]; ^2^. Boydston et al., 2001 [9]; ^3^. Drea et al., 1996 [10]; ^4^. Nolfo et al., 2022 [23]; ^5^. Casetta et al., 2022 [25].

## Data Availability

No new data were generated or analyzed in this study.

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
