# Peer review of "The Complementary Role of Gestures in Spotted Hyena (Crocuta crocuta) Communication"

_animals, 2025, doi:10.3390/ani15101366_

Round 1
Reviewer 1 Report
Comments and Suggestions for Authors
General report: The submitted manuscript titled “The Role of Vision in Spotted Hyena (Crocuta crocuta) Social Communication and Cognition” is an informative compilation of research on captive hyenas provides insight into potential behavioral innovations related to communication. In this manuscript authors discussed the Spotted hyenas live in fission-fusion societies, requiring flexible multimodal communication. Acoustic and olfactory signals have been studied more than visual communication, especially in wild populations. How ever a total 46 review articles examined social communication and cognition in wild and captive hyenas. Out of them 54% of studies focused on acoustic communication, 32% on olfaction, and only 14% on vision. 82% of research studied wild populations, leaving a knowledge gap in captive hyenas. Captive hyenas may have greater potential for behavioral innovations due to their robust social cognition.
Some critical observations: It seems like the authors are tackling a valuable topic, but the manuscript may lack clarity and organization. A well-structured paper enhances readability and ensures key points are effectively conveyed. To improve, they might focus on:
Clear sectioning: Defining distinct sections like introduction, methodology, results, and discussion. While collecting information must use PRISMA checklist and flow chart. Work on some meta-analysis. Make some objectives and hypothesis.
Logical flow: Ensuring smooth transitions between ideas to maintain coherence.
Balanced emphasis: Addressing all aspects—acoustic, olfactory, and visual communication—with proportional depth.
Stronger conclusions: Summarizing findings clearly and emphasizing their significance.
Reviewer 2 Report
Comments and Suggestions for Authors
This manuscript systematically reviews visual communication in spotted hyenas, with particular emphasis on comparative analyses between captive and wild populations. While the study provides novel insights into the adaptive value of gestural communication in this species, several critical conceptual and structural issues require clarification to strengthen the theoretical framework.
Major Concerns:
Conceptual Ambiguity in Signal Classification
The manuscript conflates "visual signals" with "gestural communication". While gestural communication (e.g., facial/body movements) constitutes a subset of visual signaling, the latter encompasses broader modalities such as static morphological displays (e.g., piloerection, body posture) and chromatic changes. We recommend explicitly distinguishing these categories and consistently using "gestural communication" when referring to dynamic body movements.
Underdeveloped Linkage to Social Structure
As a fission-fusion species, spotted hyenas' communication systems likely exhibit specialized adaptations to fluctuating group dynamics. The discussion would benefit from addressing:
How visual/gestural signals facilitate subgroup coordination during fusion events, paralleling chimpanzees' use of pant-hoot vocalizations (Wilson et al., 2001)
Whether specific gestures correlate with subgroup size/composition changes in wild populations
Line 75-90. I am not sure whether this part is necessary to a review article.
Line91-155. The logical structure of this section appears somewhat lacking in coherence. While it is beneficial to first introduce the definition of gestural communication, the subsequent paragraphs would profit from a more systematic approach. Specifically, after establishing the foundational concept, I had anticipated encountering detailed descriptions of spotted hyenas' gestural repertoire—akin to the categorizations presented in Table 1. However, the authors allocate considerable emphasis to discussing the functional aspects of these gestures rather than their specific manifestations. My recommendation would be to initially delineate the distinct gestures observed in hyenas, followed by a functional organization of these behaviors into categories such as long-distance communication, social rank signaling, emotional expression, and social cognitive processes. This dual-tiered structure—first describing behaviors then analyzing their purposes—would enhance both clarity and analytical depth.
Line 200-213: These two paragraphs primarily contrast olfactory and visual signaling from a conceptual perspective. The advantages and disadvantages of olfactory communication described here may generalize across numerous species.
I agree with the authors' assertion that "gestures in the spotted hyena repertoire serve as redundant signals." To strengthen this section's focus, I suggest revising the paragraphs to emphasize how visual signals specifically complement olfactory communication by providing supplementary information. This revised approach should first establish the foundational role of olfactory cues in conveying certain types of information—such as reproductive status or territorial markers—before demonstrating how visual gestures augment this system. For instance, olfactory signals may effectively broadcast enduring physiological states over time, while visual displays could convey immediate behavioral intentions or transient emotional states that require rapid, context-specific interpretation. This dual-signal framework would also better align with the authors' characterization of gestural redundancy, demonstrating how visual and olfactory systems interact to enhance communication fidelity in complex social environments.
Line 240-279: It is interesting to observe that captive spotted hyenas exhibit greater behavioral diversity than the wild, as this pattern typically reverses in many species under controlled conditions. This unexpected finding may indeed correlate with the species' fission-fusion social structure, where temporary subgroup formations ("fusion" events) facilitate information exchange and resource sharing in natural environments. In captivity, however, the frequency and complexity of these fusion opportunities become artificially constrained due to limited spatial dynamics and enforced group stability. Consequently, individuals may compensate by developing a broader repertoire of communicative behaviors to convey nuanced social information within the restricted context of persistent group proximity.
Please provide details about the captive situations, for example, group size/sex ratio.
Line 373: Please check the references. Scientific name of species should be italicized.
Reviewer 3 Report
Comments and Suggestions for Authors
The manuscript titled “The Role of Vision in Spotted Hyena (Crocuta crocuta) Social Communication and Cognition” provides a timely synthesis of spotted hyena communication, with a compelling argument for prioritizing visual modality research. The captive vs. wild comparison is particularly innovative and merits further exploration.
There are areas that require attention to improve clarity, accuracy, and scientific rigor. Below are some the major and minor comments for revision:
Major Comments
- The claim that spotted hyenas possess “unparalleled social intelligence” (line 73) is bold consider referencing comparative cognitive studies more directly or softening the phrasing
- Some generalizations (e.g., “resources are relatively abundant [line 235]” in captivity) should be tempered or qualified — not all zoos are equal
- Define technical terms (e.g., neophobia, altricial cubs, redundant signals) upon first use to enhance accessibility for non-specialist readers.
- Consider including a PRISMA-style flow diagram or table summarizing the search and categorization process for clarity
Minor comments:
- Line 9: “relative social status” is vague; consider specifying “dominance rank” to align with hyena social hierarchy literature.
- Line 16: Rephrase “primed for behavioral innovation” — this metaphor may not translate clearly to all readers.
- Line 41: The phrase “signals that vary in spatial and temporal scale and variable social contexts” is redundant — consider streamlining.
- Line 45–48: Clarify what “duration in a clan” entails — is it a linear progression or subject to social negotiation?
- Line 137–147: Well-described, but the “ROM” discussion is repetitive — once its role in play is established, avoid reiteration.
- Line 161: “adjusting for environmental variability” — clarify what these adjustments entail
- Line 204-209: The statement that visual signals are “less prone to eavesdropping” is compelling; cite supporting literature here, or elaborate on the theoretical basis
- Line 254–256: Consider rephrasing for clarity — it’s not entirely clear whether “successful” refers to task completion or exploratory behavior.
- The discussion of “third-party relationships” (line 328) is excellent — highlight this earlier in the introduction as part of why hyena cognition is significant.
Reviewer 4 Report
Comments and Suggestions for Authors
Dear Authors,
The manuscript certainly touches upon an interesting topic. Hyenas are social predators. The role of the senses in their lives is decisive. From the beginning to the end of the manuscript, visual perception is compared with acoustic perception and olfactory senses. But this is not taken into account in the title of the manuscript. The manuscript considers the organ of vision in a much broader context in conjunction with other senses. The authors still do not come to a consensus on what is more important and what is secondary in the lives of hyenas. This needs to be understood and their own point of view needs to be stated. Therefore, the manuscript cannot be published in its current form. The text of the manuscript is mixed up in different chapters and should be moved to the appropriate chapters. Little attention is paid to the reaction speed of the hyena. The visual material needs to be improved. Photos of hyenas with different gestures should be added. In many methodological aspects, the authors omit important information. It should be added. The comparative part of the discussion needs to be expanded and additional sources of literature on other similar predators should be cited. The focus should be on writing the conclusion of the manuscript based on the conducted research according to the presented hypothesis. There is none yet. After all comments have been addressed, the manuscript can be reviewed again.

Round 2
Reviewer 1 Report
Comments and Suggestions for Authors
Add a graphics of communication by Spotted Hyena
Reviewer 3 Report
Comments and Suggestions for Authors
No comments
Reviewer 4 Report
Comments and Suggestions for Authors
Dear Authors,
I have seen the correction of the manuscript. Additional data have been added to the manuscript. Indeed, the role of gestural communication in maintaining social cohesion in hyenas is vital, as evidenced by the retention and subsequent renewal of many gestures in captive populations, as well as the use of gestures by the captive population to assist in joint tasks. The authors have summarized the data available in the literature, essentially acting as analysts. Yes, such articles do appear in scientific journals. The review of the research results was chosen accordingly. The article takes into account the comments on the methodology. The analysis and conclusion for each chapter are sufficient and do not raise objections. References to sources of literature have been corrected.